# Mitochondrial Genomes of *Hestina persimilis* and *Hestinalis nama* (Lepidoptera, Nymphalidae): Genome Description and Phylogenetic Implications

**DOI:** 10.3390/insects12080754

**Published:** 2021-08-20

**Authors:** Yupeng Wu, Hui Fang, Jiping Wen, Juping Wang, Tianwen Cao, Bo He

**Affiliations:** 1School of Environmental Science and Engineering, Taiyuan University of Science and Technology, Taiyuan 030024, China; Fanghui0622fh@163.com; 2College of Plant Protection, Shanxi Agricultural University, Taiyuan 030031, China; wjp6269316@163.com (J.W.); jp76@163.com (J.W.); 3Department of Horticulture, Taiyuan University, Taiyuan 030012, China; 4College of Life Sciences, Anhui Normal University, Wuhu 241000, China; hebo90@ahnu.edu.cn

**Keywords:** lepidoptera, Nymphalidae, mitochondrial genome, phylogeny

## Abstract

**Simple Summary:**

In this study, the mitogenomes of *Hestina persimilis* and *Hestinalis nama* were obtained via sanger sequencing. Compared with other mitogenomes of Apaturinae butterflies, conclusions can be made that the mitogenomes of *Hestina persimilis* and *Hestinalis nama* are highly conservative. The phylogenetic trees build upon mitogenomic data showing that the relationships among Nymphalidae are similar to previous studies. *Hestinalis*
*nama* is apart from *Hestina*, and closely related to Apatura, forming a monophyletic clade.

**Abstract:**

In this study, the complete mitochondrial genomes (mitogenomes) of *Hestina persimilis* and *Hestinalis nama* (Nymphalidae: Apaturinae) were acquired. The mitogenomes of *H. persimilis* and *H. nama* are 15,252 bp and 15,208 bp in length, respectively. These two mitogenomes have the typical composition, including 37 genes and a control region. The start codons of the protein-coding genes (PCGs) in the two mitogenomes are the typical codon pattern ATN, except CGA in the *cox1* gene. Twenty-one tRNA genes show a typical clover leaf structure, however, *trnS1*(AGN) lacks the dihydrouridine (DHU) stem. The secondary structures of *rrnL* and *rrnS* of two species were predicted, and there are several new stem loops near the 5′ of *rrnL* secondary structure. Based on comparative genomic analysis, four similar conservative structures can be found in the control regions of these two mitogenomes. The phylogenetic analyses were performed on mitogenomes of Nymphalidae. The phylogenetic trees show that the relationships among Nymphalidae are generally identical to previous studies, as follows: Libytheinae\Danainae + ((Calinaginae + Satyrinae) + Danainae\Libytheinae + ((Heliconiinae + Limenitidinae) + (Nymphalinae + (Apaturinae + Biblidinae)))). *Hestinalis*
*nama* is apart from *Hestina*, and closely related to Apatura, forming monophyly.

## 1. Introduction

*Hestina persimilis* and *Hestinalis nama* belong to the lepidopteran family Nymphalidae, Apaturinae and mainly distribute in the Palaearctic and Oriental region. Their adults inhabit mountainous broad-leaved forests and present the habit of sipping tree juice and water in wetlands. The larvae were reported as a kind of agriculture pest of the host plants, Ulmaceae. At present, there are only one species (*H. nama*) of the genus *Hestinalis* and three species (including *Hestina persimilis*, *Hestina assimilis* and *Hestina nicevillei*) of the genus *Hestina* distributed in China. *Hestinalis nama* was originally described as *Diadema nama* by Doubleday in 1844 [1]. However, in subsequent studies, latter scholars placed it under the genus *Hestina* [2,3]. The classification of butterflies is mainly based on the characteristics of external genitalia and wing veins. Morphologically, including genitalic structure, *Hestina* and *Hestinalis* are easily separable [4]. In recent studies, *Hestinalis* was treated as a distinct genus [5]. Although most modern literature chooses to separate them, some literature, Wu and Hsu still treats them as one [6]. In this paper, phylogenetic analysis shows that *Hestinalis nama* is apart from *Hestina*. Therefore, we also separate them apart.

Mitogenome fragments have been extensively used in phylogenetic analysis for butterflies and moths, particularly for the *cox1* gene which was primarily used as a DNA barcoding for animals [7,8,9,10,11]. In the BOLD system [12], lepidopteran insects consist of the largest amount of data being sequenced. However, the phylogenetic relationships at different taxonomic levels are still controversial [13,14,15]. It has been proposed that mitochondrial genomes might provide more genetic information than a single gene fragment [16,17,18]. Therefore, sequencing more mitogenomes might improve our understanding of evolution and phylogeny at different taxonomic levels in Lepidoptera. Furthermore, the mitogenome has been widely used in the areas of population genetic structure, gene drift and phylogenetics, because of its characteristics of maternal inheritance, small genome size (15–20 kb in length) and rapid rate of evolution [19,20]. To date, only one complete mitogenome (*H. assimilis*) has been sequenced from the genus *Hestina*; other species and *Hestinalis nama* were all not sequenced, which is quite limited and will restrict our understanding of evolution in Nymphalidae at the genomic level. In this study, the mitogenomes of *H. persimilis* and *H. nama* were obtained, with the aim of: (1) providing a comparative analysis of Apaturinae mitogenomes, including nucleotide composition, codon usage, gene arrangement, prediction of tRNA and rRNA secondary structures and novel features of the control region, and (2) reconstructing the phylogenetic relationships among subfamilies in Nymphalidae based on mitogenomes data. 

## 2. Materials and Methods

### 2.1. Sampling and DNA Sequencing

Specimens of *H. persimilis* and *H. nama* were collected from the Sichuan and Yunnan Provinces of China in 2010. Specimens were being made, followed by morphological identification. One side of the hindfoot for each sample was preserved in absolute ethanol and stored in −20 °C freezer in College of Plant Protection, Shanxi Agricultural University, Taiyuan, China.

The DNA extraction kit and primers [21] (Appendix A) were produced by Shanghai Major Biomedical Technology Co., Ltd. (Shanghai, China). The reaction systems of PCR amplifications were 25 μL, including upstream and downstream primers 0.5 μL, respectively, PCR Master Mix 12.5 μL, DNA template 3 μL, and ddH_2_O 8.5 μL. The amplification reaction conditions were as follows: initial denaturation at 94 °C for 2 min; 35 cycles of denaturation at 94 °C for 1 min, annealing at 53 °C for 45 s, extension at 72 °C for 1 min, and a final extension step at 72 °C for 4 min. PCR products were detected by 1% agarose gel electrophoresis. All the gene fragments were sent to Shanghai Major Biomedical Technology for sequencing.

### 2.2. Annotation and Analysis of Mitochondrial DNA

The original sequence fragments were assembled with SeqMan (Steve ShearDown, 1998–2001 version reserved by DNASTAR Inc., Madison, WI, USA) to get a complete mitogenome. The secondary structure of tRNA genes were determined by tRNAscan-SE Search Server (http://lowelab.ucsc.edu/tRNAscan-SE/; accessed on 28 June 2021) [22]. Putative tRNA genes, including *trnH* and *trnS1*(AGN), which could not be found by tRNAscanSE, were confirmed by comparison with the homologous genes of other Apaturinae species. PCGs and rRNA genes were identified by the MITOS webserver with invertebrate genetic code [23]. The nucleotide composition and codon usage of PCGs were calculated with MEGA-X [24]. Determination of tandem repeat sequences in control regions were performed using the Tandem Repeats Finder online software (http://tandem.bu.edu/trf/trf.html; accessed on 10 May 2020) [25]. The mitogenomes of *H. persimilis* and *H. nama* were uploaded to GenBank, with the accession numbers of MT110153 and MT110154.

### 2.3. Phylogenetic Analysis

Phylogenetic analysis was performed on the dataset of 13 PCGs from 54 complete or nearly complete mitogenomes of Nymphalidae, with two Papilionidae species selected as outgroups (Appendix A). All assembled PCGs of 56 mitogenomes were aligned through MEGA-X. The optimal partition tactics and substitution models were selected by PartitionFinder v2 (Appendix A) [26,27,28]. The maximum likelihood (ML) and Bayesian inference (BI) analyses were conducted through the online CIPRES Science Gateway [29]. The ML analysis was performed with RAxML-HPC2 on XSEDE [30], with GTRGAMMA model applied to all partitions. Bootstrap values were estimated with 1000 replicates. The BI analyses were carried out through two independent Markov chain Monte Carlo (MCMC) chains, which were set for 1,000,000 generations, with sampling per 1000 generations.

## 3. Results and Discussion

### 3.1. Mitogenomes Organization

The complete mitogenomes of *H. persimilis* and *H. nama* are 15,252 and 15,208 bp. They share the consistent gene organization, order and arrangement with most of other lepidopterans, including 13 PCGs, 22 tRNAs and 2 rRNAs (*rrnL* and *rrnS*) (Figure 1 and Figure 2). The mitogenome is circular with two strands. The heavy strand (H-strand) encodes most of the genes (9 PCGs and 14 tRNAs), while the light strand (L-strand) contains the remaining reverse complementary genes (four PCGs, eight tRNAs and two rRNAs), as shown in Table 1 and Table 2. In addition, the nucleotide composition of the two species are both AT-biased, similar to other lepidopterans. The AT contents of the mitogenomes of *H. persimilis* and *H. nama* are 80.9 and 79.2%, respectively (Table 3 and Table 4). The obvious AT-biased (Appendix A) is generally believed to be related to the evolution of mitochondrial origin [31].

### 3.2. Protein Coding Genes and Codon Usage

Orthologs from the two *Hestina* mitogenomes present similar start and stop codons. Most PCGs start with the typical initial codon ATN, but *cox1* initiates with CGA. In particular, the putative start codon CGA in the *cox1* gene is a common feature of most sequenced lepidopterans, but a few species start with codon ATG, ATT, ATA or TTG. While most PCGs end with the stop codon TAA or TAG, truncated codon T is also detected in *cox2* and *nad4*. It has been proposed that truncated stop codons can be completed by polyadenylation, which was also found in other insectan mitogenomes [32]. 

Relative synonymous codon usage (RSCU) can directly reflect the preference of codon usage (Appendix A). The total number of codons of 13 PCGs are 3703 for *H. persimilis* and 3709 for *H. nama*. The RSCU of twelve Apaturinae species show the same codon preference pattern (Figure 3). The mainly used codon families are *Leu1* (CUN), *Ile*, *Phe* and *Met*. There are at least 75 codons (CDs) per thousand CDs in each of them (Figure 4), among which *Leu1* has the highest utilization rate. Relative synonymous codon usage (RSCU) of Apaturinae show that degeneration codons are biased to use more A/T than G/C. The six most prevalent codons in Apaturinae, including AUU (I), AUA (M), AAU (N), UUU (F), UUA (L) and UAU (Y), are all composed of A and/or T. Conversely, some GC-rich codons are seldom utilized in the Apaturinae species. For example, the codon UCG, CCG are not used in *H. persimilis*, while CUG, GUC and CCG are absent in *Sasakia charonda*. This phenomenon is common among lepidopterans [33,34], which indicates that the GC content of genes is closely related to codon preference [35,36].

### 3.3. Transfer RNAs and Ribosomal RNAs

All 22 tRNAs typical of lepidopteran mitogenomes are found in the mitogenomes. Most tRNA genes are in classical clover-leaf secondary structures except for *trnS1*(AGN), with its DHU arm forming a simple loop, which is considered as a typical feature in metazoan mitogenomes (Figure 5) [37]. Additionally, the anticodon stem of *trnS1*(AGN) may be shortened as of base mismatch in some insect mitogenomes [38]. Previous studies showed that not only *trnS1*(AGN), but also some other tRNAs, such as *trnS2*(UCN) and *trnG*, lack a DHU or TΨC arm [39]. Missing a DHU arm and base mismatch are thermodynamically unstable, which indicate that a DHU arm might not really exist. Accordingly, this special structure of *trnS1*(AGN) still needs further investigation. In addition, it has been shown that some isoforms of tRNAs can be found in control regions or some PCGs on the L-strand of mitogenomes. The isoforms of tRNAs can also be folded into cloverleaf structures. However, it is not clear whether their functions are similar to those of tRNAs [40] or not. 

The *rrnL* gene is found between *trnL* (CUN) and *trnV*, while the *rrnS* gene is located between *trnV* and the control region. The lengths of *rrnL* genes of *H. persimilis* and *H. nama* mitogenomes are 1334 bp (AT content 84.29%) and 1327 bp (AT content 83.44%). The sizes of *rrnS* genes of *H. persimilis* and *H. nama* are 776 bp (AT content 85.03%) and 774 bp (AT content 84.39%). The secondary structure of *rrnL* genes include six structural domains except for that domain III is absent in arthropods (Figure 6). The *rrnS* genes include three structural domains (Figure 7). Both the secondary structures of *rrnL* and *rrnS* of the two species are roughly similar to other lepidopterans, such as *Amata emma*, *Apis mellifera* [41], *Grapholita molesta* [42], *Manduca sexta* [43], etc. The microsatellite sequence (TA)_n_ is not found in *rrnL* and *rrnS*, but exists in other insects (e.g., *Choristoneura longicellana*). There are several new stem loops near the 5′ of *rrnL* secondary structure, and these loops were not found in other insects.

Dashes, black dots and circles indicate the Watson-Crick base pairings, G-U bonds and U-U, A-A, A-C and A-G bonds, respectively.

### 3.4. Intergenic and Overlapping Regions

It has been proposed that mitogenomes tend to be highly economized in size by eliminating or reducing intergenic spacers [44]. However, by excluding the control region, 12 intergenic spacers (1 to 91 bp, 150 bp in total) are found in *H. persimilis*, and 11 intergenic spacers (1 to 69 bp, 118 bp in total) are found in *H. nama*. It has been reported that Lepidopteran mitogenomes usually have two typical and relatively conservative intergenic spacers. The longer one is located between *trnQ* and *nad2* genes, with the length of 91 and 69 bp in *H. persimilis* and *H. nama*. Previous studies found that the nucleotide sequence of the *trnQ-nad2* spacer and the *nad2* gene have a highly similarity. It has been inferred that the *trnQ-nad2* spacer may come from the *nad2* gene [45]. The other shorter spacer is located between *trnS2*(UCN) and *nad1* genes, with the length of 22 and 13 bp in *H. persimilis* and *H. nama*, respectively, sharing a conserved sequence of ATACTAA.

Comparing with the intergenic spacers, the overlapping regions are more conservative [46]. Fourteen overlapping spacers (1 to 26 bp, 66 bp in total) are found in *H. persimilis*, and fourteen overlapping spacers (1 to 8 bp, 42 bp in total) are found in *H. nama*. ATP8 and ATP6 overlap with the ATGATAA motif in the two mitogenomes, which had also been reported in many other lepidopterans [47].

### 3.5. Putative Control Regions

The control region, also known as the A + T-region or D-loop, is always the largest intergenic spacer in animal mitogenomes and considered as the initial region for replication [48]. The control regions (376 bp in *H. persimilis* and 390 bp in *H. nama*) in the two mitogenomes are located between *rrnS* and *trnM*. The AT content is also the highest in mitogenomes (91.23% in *H. persimilis* and 88.72% in *H. nama*). Previous studies indicated that the control region is the segment with fastest evolutionary rate and can be used as an important molecular marker for animal population genetics. 

There are generally four conserved structures in the control region, including a motif of ATAGA located at downstream of *rrnS* followed by 19 bp Poly-T stretch, a poly-A stretches (9 bp in *H. persimilis* and 6 bp in *H. nama*) at the upstream of *trnM* (Figure 8 and Figure 9), the microsatellite-like repeat regions ((AT)_10_ in *H. persimilis* and (AT)_6_ in *H. nama*), and the repeated sequences (23 bp in *H. persimilis* and 25 bp in *H. nama*). All these characteristics are generally considered to be related to the transcription or replication of mitogenomes [49]. Although the location of initial replication region in complete metamorphosis insects (including lepidopterans) are different, they all located after polyT (about 10–20 bp) (Figure 10) [50]. Accordingly, polyT may be involved in the recognition of the initial replication region [51].

### 3.6. Phylogenetic Analysis

The phylogenetic analyses are performed on concatenated nucleotide sequences of 13 PCGs (data matrix 10,472 bp) derived from 54 available Nymphalidae mitogenomes, with two Papilionidae species serving as outgroups. These 54 sequences represent 9 subfamilies: Apaturinae; Biblidinae; Calinaginae; Danainae; Heliconiinae; Libytheinae; Limenitidinae; Nymphalinae and Satyrinae. The BI (Figure 11) and ML trees (Figure 12) have roughly the same topology, except for Danainae and Libytheinae, which are in different locations. The overall relationship is generally as follows: Libytheinae\Danainae + ((Calinaginae + Satyrinae) + Danainae\Libytheinae + ((Heliconiinae + Limenitidinae) + (Nymphalinae + (Apaturinae + Biblidinae)))). Earlier, Heliconiinae, Limenitidinae, Biblidinae and Apaturinae were considered as “core nymphalids” [52,53]. Subsequently, there are ten widely recognized subfamilies in Nymphalidae, including Apaturinae, Libytheinae, Danainae, Morphinae, Satyrinae, Calinaginae, Eliconiinae, Limenitidinae, Charaxinae and Nymphalinae [54]. Although mitogenomes of the subfamilies Morphinae, Eliconiinae and Charaxinae are not yet available, the topology is generally identical to those of other studies [55,56,57]. *Hestina* and *Euripus*, *Sasakia* come together to form a branch; *Apatura* and *Hestinalis* form a branch; *Chitoria*, *Timelaea* and *Herona* form a branch. Early on, *Apatura* included *Chitoria* [58], while in this paper, *Chitoria* is distant from *Apatura*. Ohshima et al. (2010) has published an updated work on Apaturinae butterflies, considering that *Chitoria* should been removed from *Apatura* [59], their grouping is concordant to this paper. *H. persimilis*, *H. assimilis* and the genus *Euripus* (represented by *Euripus nyctelius*) form a branch, of which the result is the same as the morphological study. *Hestinalis* is a clearly distinct lineage within the genus *Hestina*, not together with *H. persimilis* and *H. assimilis*, and closely related to Apatura, that is considered to be a result of mimicry [60], which is also consistent with our paper.

## 4. Conclusions

The mitogenomes of *H. persimilis* and *H. nama* were obtained using sanger sequencing. Comparing them with other mitogenomes of Apaturinae butterflies, the conclusion can be drawn that the mitogenomes are highly conserved, sharing the same gene order, gene location, codon usage, nucleotide composition and AT-biased pattern. The secondary structures of *rrnL* and *rrnS* of two species are roughly similar to other lepidopterans. Although the control regions vary greatly in length, their structure has not changed much, which includes four basic conservative regions. The topology of phylogenetic analyses are generally identical to those of other studies. *Hestinalis*
*nama* is not grouped with *Hestina*, and is closely related to Apatura, which is consistent with early studies.

## Figures and Tables

**Figure 1 insects-12-00754-f001:**
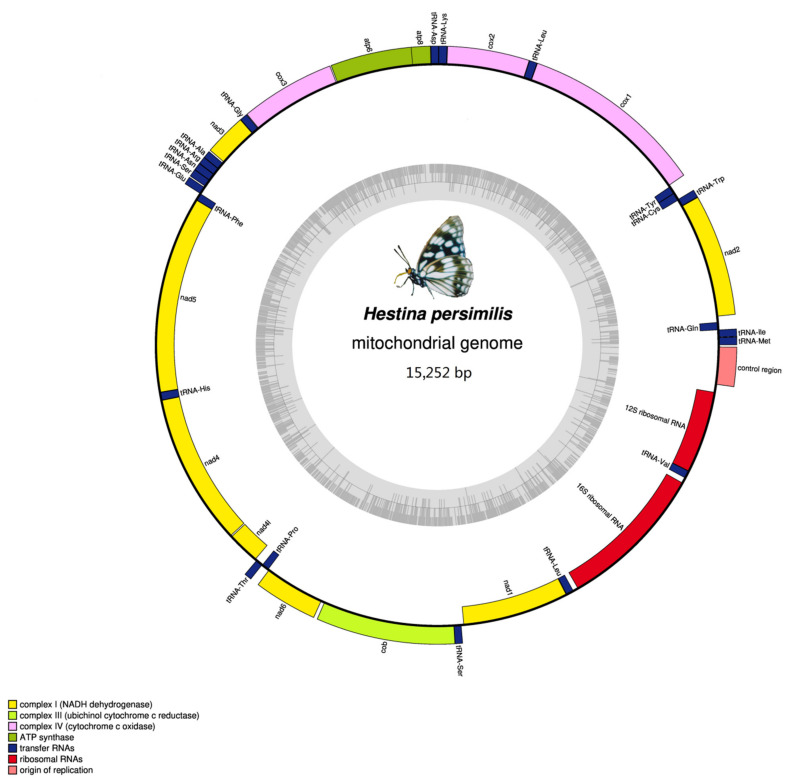
Linear map of the mitogenome of *Hestina persimilis*. The J-strand is located on the linear map, and the N-strand is under the linear map.

**Figure 2 insects-12-00754-f002:**
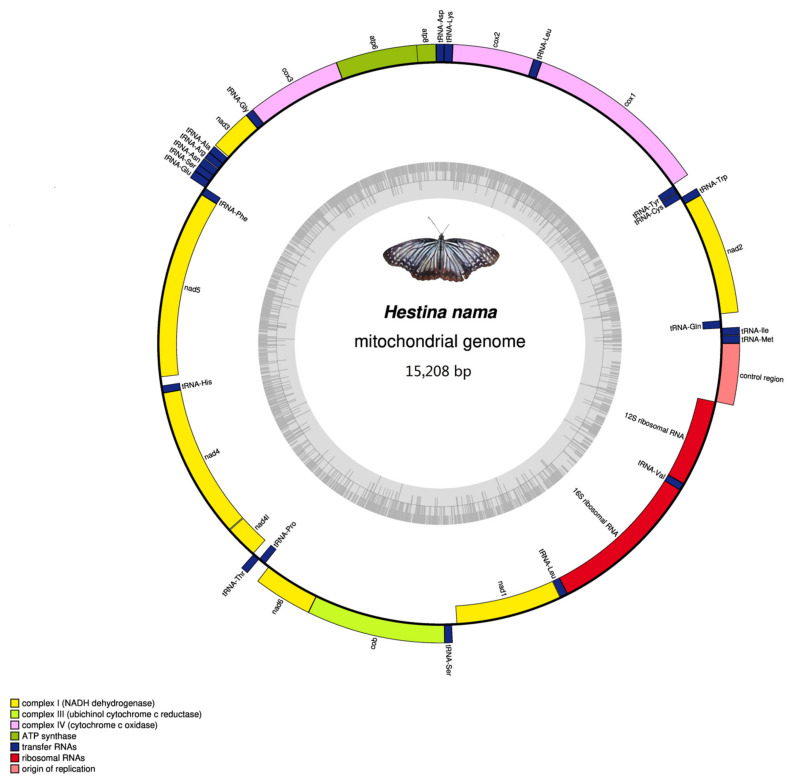
Linear map of the mitogenome of *Hestinalis*
*nama*. The J-strand is located on the linear map, and the N-strand is under the linear map.

**Figure 3 insects-12-00754-f003:**
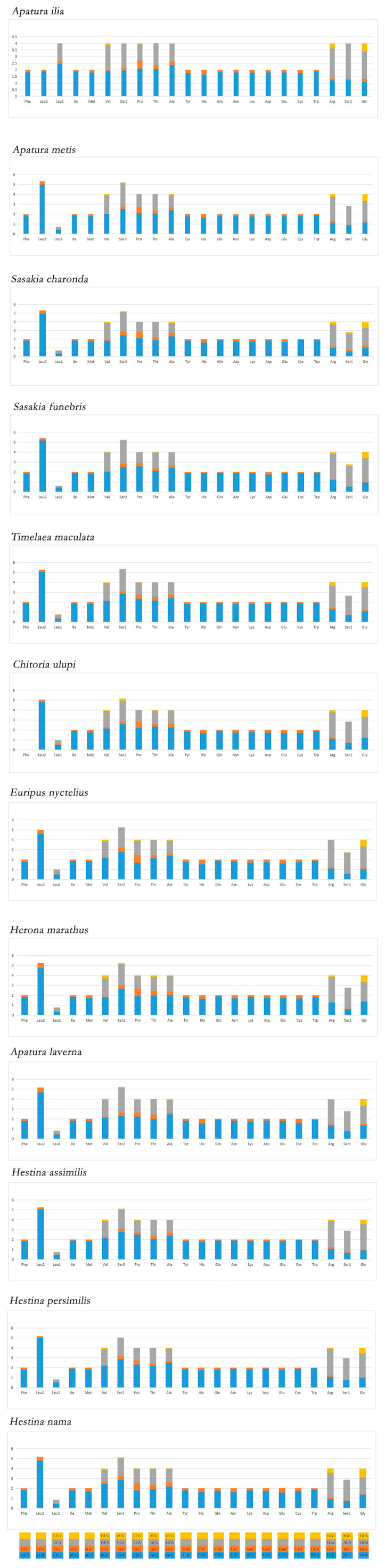
Relative Synonymous Codon Usage (RSCU) in the mitogenomes of twelve Apaturinae species. Codon families are provided on the X axis.

**Figure 4 insects-12-00754-f004:**
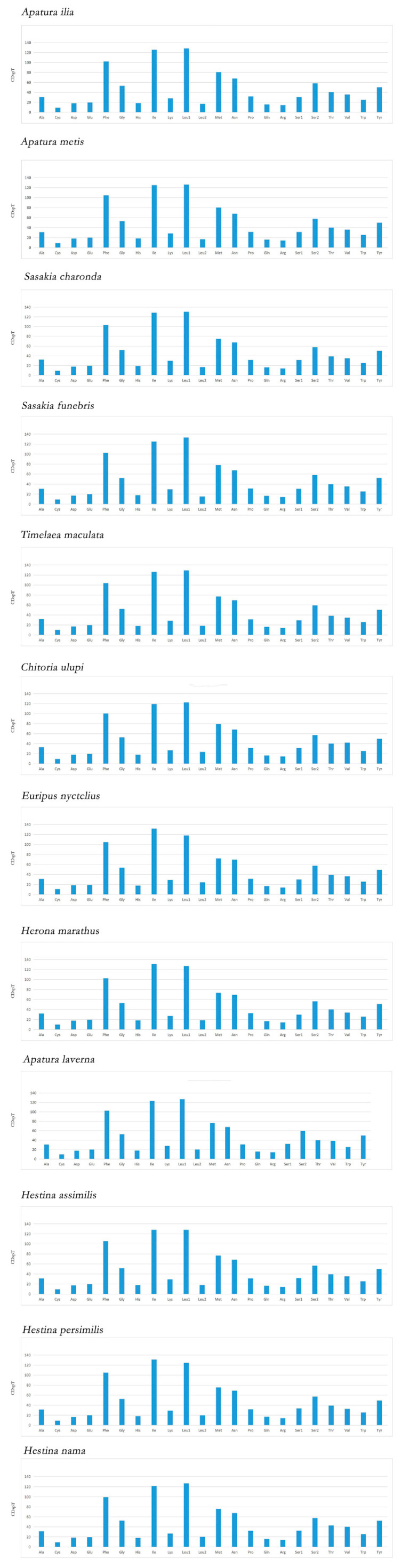
Codon distributions in the mitogenomes of twelve Apaturinae species. CDspT: codons per thousand codons.

**Figure 5 insects-12-00754-f005:**
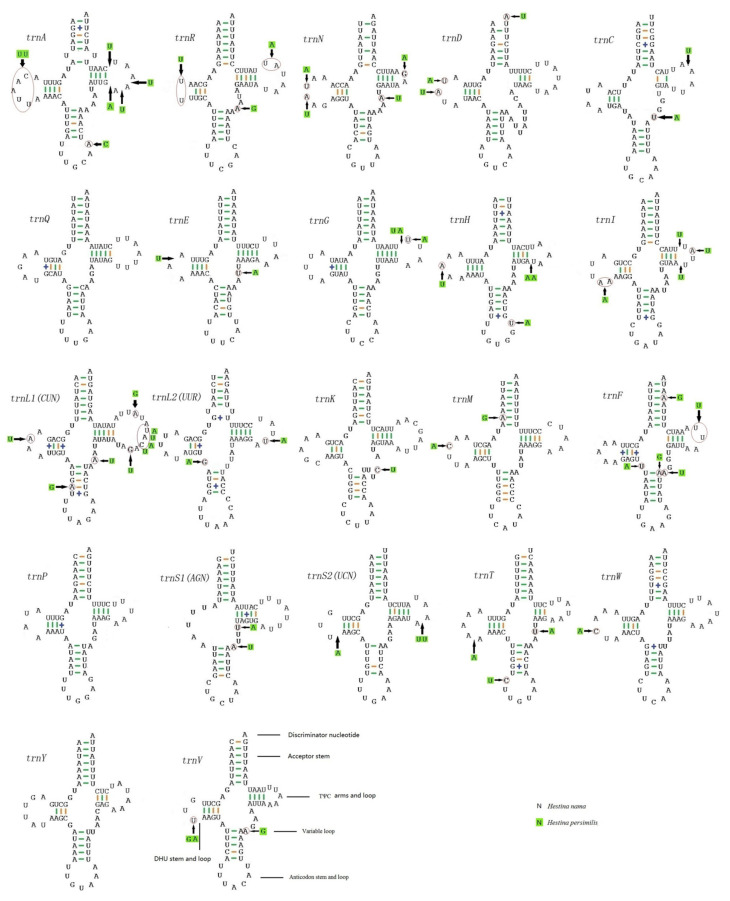
Predicted secondary cloverleaf structure for the tRNAs of *Hestina persimilis* and *Hestinalis*
*nama*. Dashes (–) and pluses (+) indicate the Watson–Crick base pairings and G-U bonds, respectively.

**Figure 6 insects-12-00754-f006:**
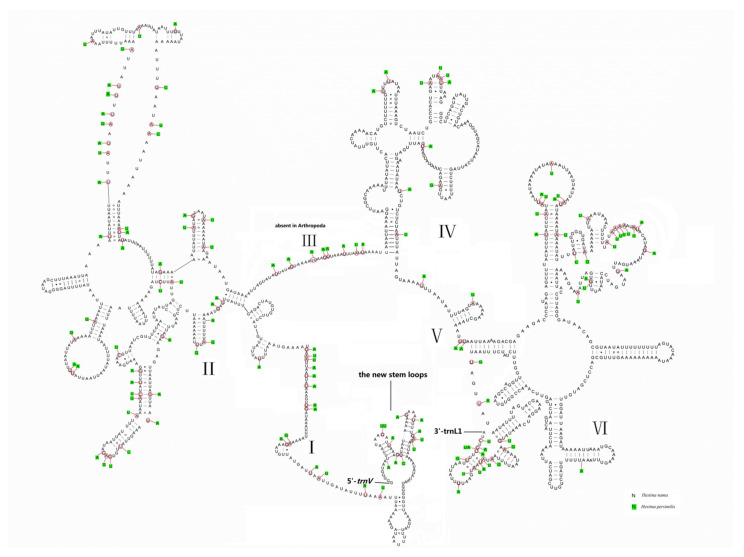
Predicted *rrnL* secondary structure of *Hestina persimilis* and *Hestinalis nama* mitogenomes.

**Figure 7 insects-12-00754-f007:**
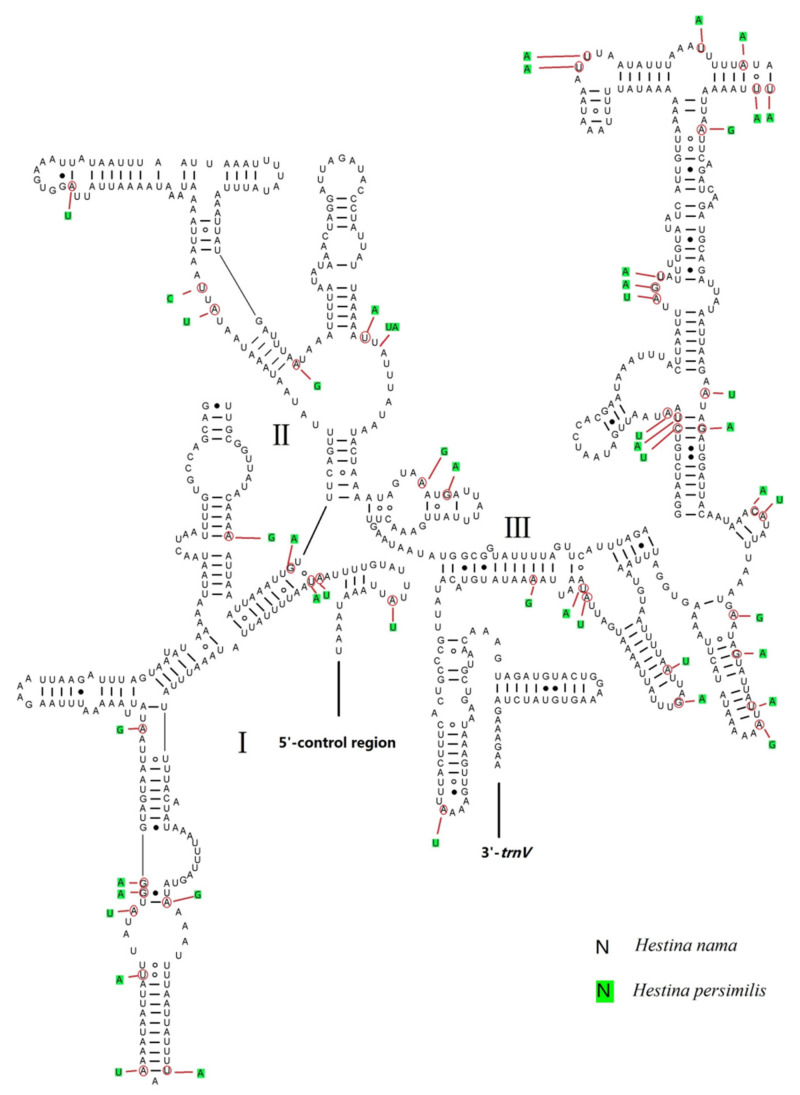
Predicted *rrnS* secondary structure of *Hestina persimilis* and *Hestinalis nama* mitogenomes. Dashes, black dots and circles indicate the Watson–Crick base pairings, G-U bonds and U-U, A-A, A-C, A-G bonds, respectively.

**Figure 8 insects-12-00754-f008:**
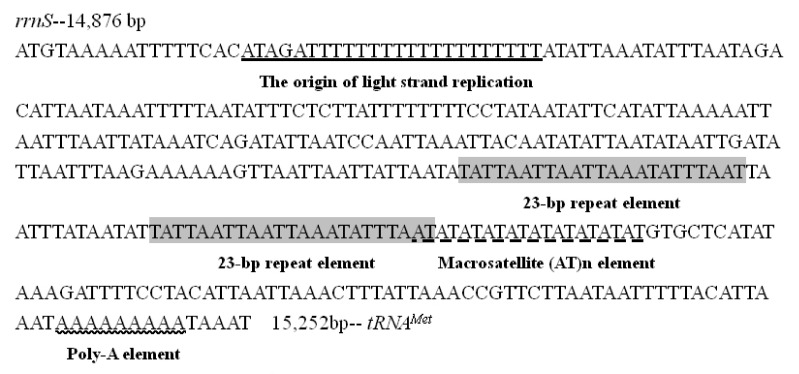
The control region in *Hestina persimilis*.

**Figure 9 insects-12-00754-f009:**
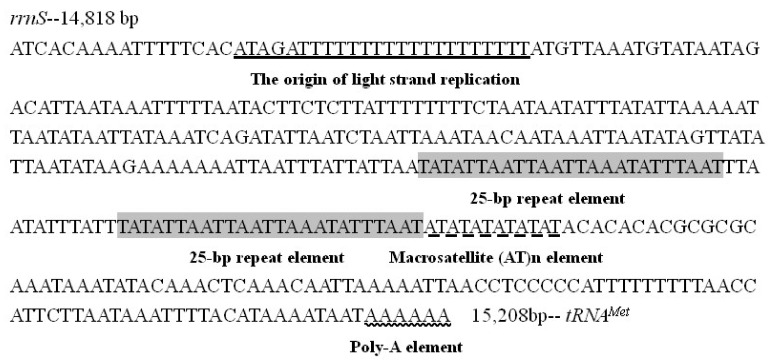
The control region in *Hestinalis*
*nama*.

**Figure 10 insects-12-00754-f010:**
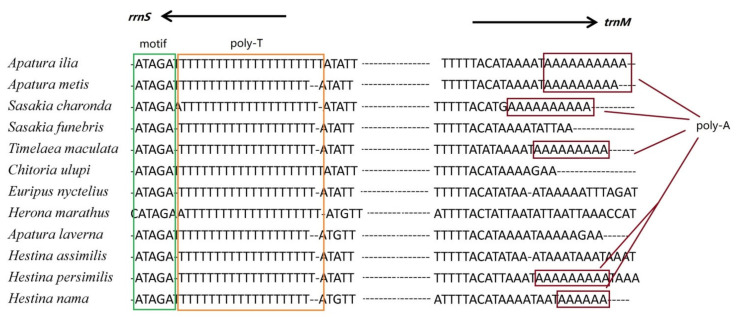
Alignment of motif, Poly(T) and poly(A) in control regions of twelve species in Apaturinae.

**Figure 11 insects-12-00754-f011:**
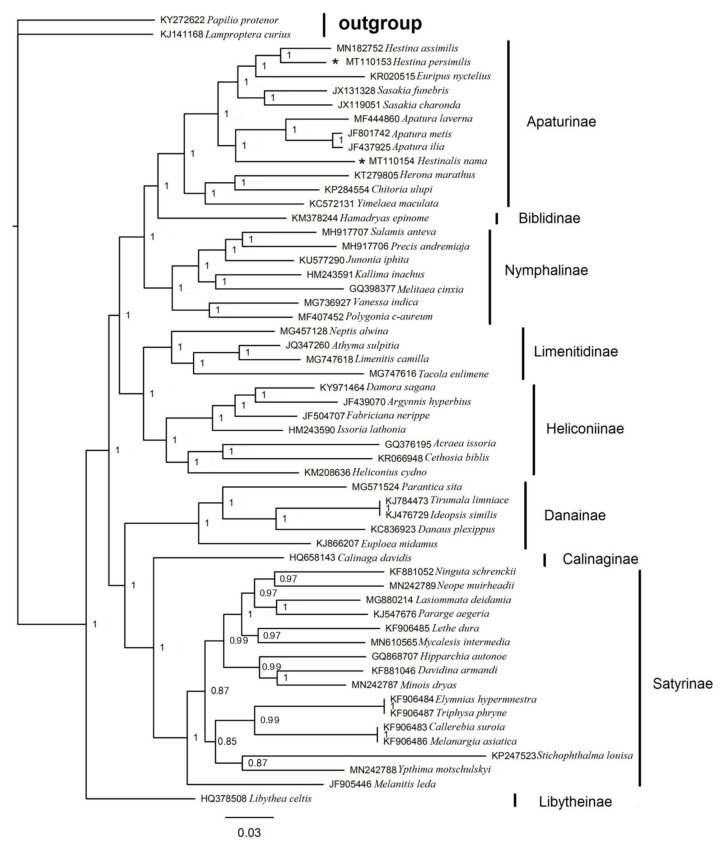
Inferred phylogenetic relationships among Lepidoptera based on the concatenated nucleotide sequences of 13 PCGs using BI. Numbers on branches are Bayesian posterior probabilities. *Papilio protenor* (KY272622) and *Lamproptera curius* (KJ141168) are used as outgroups.

**Figure 12 insects-12-00754-f012:**
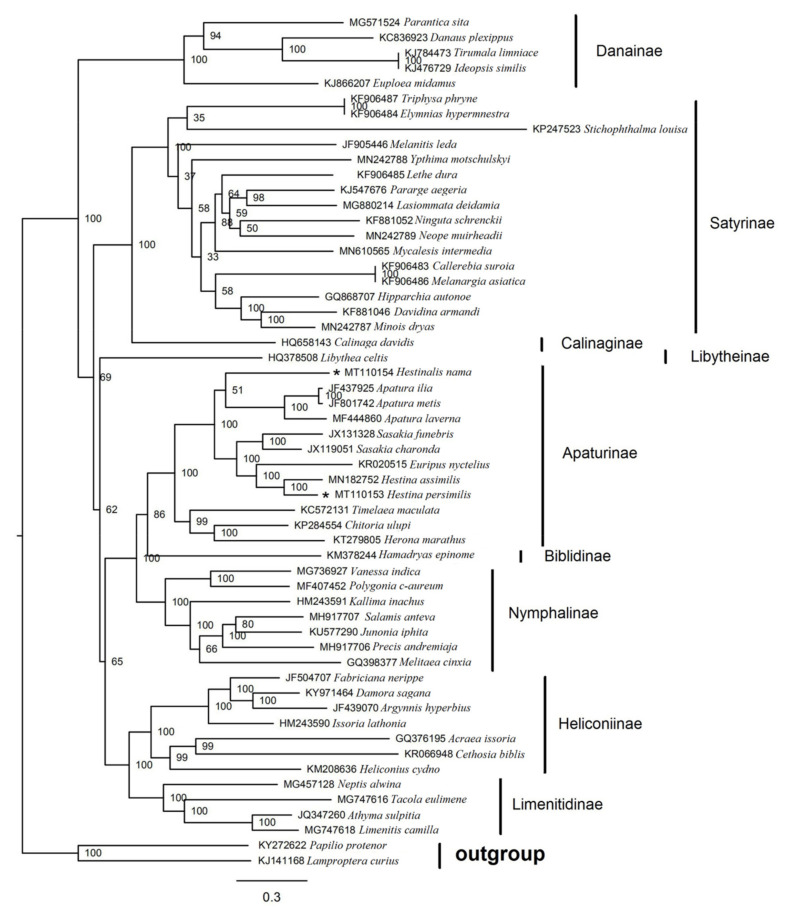
Inferred phylogenetic relationships among Lepidoptera based on the concatenated nucleotide sequences of 13PCGs using ML. Numbers on branches are bootstrap percentages. *Papilio protenor* (KY272622) and *Lamproptera curius* (KJ141168) are used as outgroups.

**Table 1 insects-12-00754-t001:** Annotation of *Hestina persimilis* mitogenome.

Gene	Direction	Location	Size	Anticodon	Start Codon	Stop Codon	Intergenic Nucleotides
*trnM*	F	1–68	68	CAT 32–34			
*trnI*	F	69–134	66	GAT 98–100			0
*trnQ*	R	132–200	69	TTG 159–161			−3
*nad2*	F	292–1305	1014		ATT	TAA	91
*trnW*	F	1304–1371	68	TCA1335–1337			−2
*trnC*	R	1364–1427	64	GCA 1397–1399			−8
*trnY*	R	1428–1492	65	GTA 1359–1461			0
*cox1*	F	1498–3033	1536		CGA	TAA	5
*trnL2* (UUR)	F	3029–3095	67	TAA 3059–3061			−5
*cox2*	F	3096–3774	679		ATG	T	0
*trnK*	F	3772–3842	71	CTT 3802–3804			−3
*trnD*	F	3842–3907	66	GTC 3872–3874			−1
*atp8*	F	3908–4069	162		ATC	TAA	0
*atp6*	F	4063–4737	675		ATG	TAA	−7
*cox3*	F	4737–5525	789		ATG	TAA	−1
*trnG*	F	5528–5594	67	TCC 5558–5560			2
*nad3*	F	5595–5948	354		ATT	TAG	0
*trnA*	F	5947–6014	68	TGC 5976–5978			−2
*trnR*	F	6014–6077	64	TCG 6040–6042			−1
*trnN*	F	6090–6155	66	GTT 6121–6123			12
*trnS1* (AGN)	F	6154–6213	60	GCT 6171–6173			−2
*trnE*	F	6216–6280	65	TTC 6245–6247			2
*trnF*	R	6279–6342	64	GAA 6310–6312			−2
*nad5*	R	6317–8077	1761		ATT	TAA	−26
*trnH*	R	8075–8141	67	GTG 8109–8111			−3
*nad4*	R	8142–9480	1339		ATG	T	0
*nad4L*	R	9482–9772	291		ATA	TAA	1
*trnT*	F	9780–9844	65	TGT 9811–9813			7
*trnP*	R	9845–9908	64	TGG 9877–9879			0
*nad6*	F	9911–10438	528		ATA	TAA	2
*cob*	F	10,442–11,593	1152		ATG	TAA	3
*trnS2* (UCN)	F	11,596–11,662	67	TGA 11,625–11,627			2
*nad1*	R	11,685–12,626	942		ATG	TAA	22
*trnL1* (CUN)	R	12,628–12,702	75	TAG 12,671–12,673			1
*rrnL*	R	12,703–14,036	1334				0
*trnV*	R	14,037–14,100	64	TAC 14,069–14,071			0
*rrnS*	R	14,101–14,876	776				0
Control region		14,877–15,252	376				0

**Table 2 insects-12-00754-t002:** Annotation of *Hestinalis*
*nama* mitogenome.

Gene	Direction	Location	Size	Anticodon	Start Codon	Stop Codon	Intergenic Nucleotides
*trnM*	F	1–68	68	CAT 32–34			
*trnI*	F	69–133	65	GAT 99–101			0
*trnQ*	R	131–199	69	TTG 158–160			−3
*nad2*	F	269–1282	1013		ATT	TAA	69
*trnW*	F	1281–1348	68	TCA 1312–1314			−2
*trnC*	R	1341–1403	63	GCA 1372–1374			−8
*trnY*	R	1404–1468	65	GTA 1435–1437			0
*cox1*	F	1474–3009	1536		CGA	TAA	5
*trnL2* (UUR)	F	3005–3071	67	TAA 3035–3037			−5
*cox2*	F	3072–3750	679		ATG	T	0
*trnK*	F	3748–3818	71	CTT 3778–3780			−3
*trnD*	F	3818–3883	66	GTC 3848–3850			−1
*atp8*	F	3884–4042	159		ATC	TAA	0
*atp6*	F	4036–4713	678		ATG	TAA	−7
*cox3*	F	4713–5501	789		ATG	TAA	−1
*trnG*	F	5504–5568	65	TCC 5534–5536			2
*nad3*	F	5566–5922	357		ATA	TAG	−3
*trnA*	F	5921–5987	67	TGC 5953–5955			−2
*trnR*	F	5987–6052	66	TCG 6014–6016			−1
*trnN*	F	6053–6118	66	GTT 6084–6086			0
*trnS1* (AGN)	F	6117–6176	60	GCT 6134–6136			−2
*trnE*	F	6180–6243	64	TTC 6108–6210			3
*trnF*	R	6244–6308	65	GAA 6276–6278			0
*nad5*	R	6308–8044	1737		ATT	TAA	−1
*trnH*	R	8042–8106	65	GTG 8071–8073			−3
*nad4*	R	8107–9445	1339		ATG	T	0
*nad4L*	R	9447–9731	285		ATG	TAA	1
*trnT*	F	9744–9807	64	TGT 9774–9776			12
*trnP*	R	9808–9871	64	TGG 9840–9842			0
*nad6*	F	9874–10401	528		ATA	TAA	2
*cob*	F	10,406–11,554	1149		ATG	TAA	4
*trnS2* (UCN)	F	11,561–11,624	64	TGA 11,589–11,591			6
*nad1*	R	11,638–12,579	942		ATG	TAA	13
*trnL1* (CUN)	R	12,581–12,654	74	TAG 12,623–12,625			1
*rrnL*	R	12,655–13,981	1327				0
*trnV*	R	13,982–14,044	63	TAC 14,014–14,016			0
*rrnS*	R	14,045–14,818	774				0
Control region		14,819–15,208	390				0

**Table 3 insects-12-00754-t003:** Base composition of *Hestina persimilis* mitogenome.

	Size (bp)	A%	T%	G%	C%	A + T%	G + C%
mtDNA	15,252	39.7	41.2	7.6	11.5	80.9	19.1
PCGs	11,222	33.8	45.9	10.4	9.9	79.7	20.3
tRNA	1459	41.7	39.7	11.0	7.7	81.4	18.7
*rrnL*	1334	44.5	39.8	10.5	5.2	84.3	15.7
*rrnS*	776	43.6	41.5	10.1	4.9	85.1	15
Control region	376	43.6	47.6	2.9	5.9	91.2	8.8

**Table 4 insects-12-00754-t004:** Base composition of *Hestinalis*
*nama* mitogenome.

	Size (bp)	A%	T%	G%	C%	A + T%	G + C%
mtDNA	15,208	39.9	39.3	7.9	12.9	79.2	20.8
PCGs	11,192	32.8	44.8	11.4	11.0	77.6	22.4
tRNA	1449	42.0	39.3	10.8	7.9	81.3	18.7
*rrnL*	1327	40.2	43.4	5.3	11.1	83.6	16.4
*rrnS*	774	41.2	43.9	5.2	9.7	85.1	14.9
Control region	390	44.9	43.8	2.6	8.7	88.7	11.3

## Data Availability

All sequences were deposited in the GenBank under accession numbers of MT110153 and MT110154.

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
