# Peer review of "Mitochondrial Genomes of Hestina persimilis and Hestinalis nama (Lepidoptera, Nymphalidae): Genome Description and Phylogenetic Implications"

_insects, 2021, doi:10.3390/insects12080754_

Round 1

Reviewer 1 Report

  1. Since authors changed their goal of this manuscript, I feel that the results could support some of their new aims, however, this manuscript became normal as other data papers. No new finding, except two new mitochondrial genomes provided. There are lots of new mitochondrial genomes published in other journals every day, I do not favor to accept your paper in this journal.
  2. Since authors are only focus on two species relationships of Apaturinae, the contribution of this work is very limited. It is very easy to produce mitochondrial genomes via next-generation sequencing now.
  3. I as a reader, do not interests in this article in Insects, but I encourage authors to submit your efforts to other places.

Reviewer 2 Report

The revised MS has partly solved the previous problems. It is very pity that the authors' lab cannot access to more samples in Hestina and Hestinalis globally, otherwise this would be a great study. However, I fully understand the constrains in obtaining foreign materials, so alternatively they can focus on the current issue, but must tune down many conclusions. I think it is acceptable after a minor revision. Please see my comments below.

1) I encourage the authors to edit the title and make it more specific, Apaturinae is too big, only two species cannot justify. Please consider "Mitochondrial Genomes of Hestina persimilis and Hestinalis nama: Genome Description and Phylogenetic Implications".

2) Line 11: persimilis and nama are now in two genera, please use full genus names for them, don't abbreviate, this may cause confusion.

3) Line 14: Change 'identical' to 'similar', nothing can be identical in such case.

4) Lines 34-38: Please edit the text, start and focus on Hestinalis nama and Hestina persimilis, making the scientific ground more solid. The current narrative is too loose.

5) Ling 39: Please cite Doubleday E. (1844), I remember I gave you the reference in my last comments. "Diadema nama by Doubleday in 1844, on page 97 of the first volume of “List of the Specimens of Lepidopterous insects in The Collection of the British Museum”", you only need to check other details of this literature, such as number of pages, publisher, and place of publication.

6) Line 40 and Ref. 2: Chou I., not Chou Y. Again, please use the authos' choice of name, not always Hanyu Pinyin.

7) Lines 41-45: I believe here is a good place to say different voice regarding Hestina and Hestinalis, although most modern literature choose to separate them, some literature, sensu Wu & Hsu (2017) still treats them as one. Use different views to build your own point.

8) Line 139 and Lines 169-180: I think the parentheses denoting the codons after tRNA genes should not in italic font. Please also check the rest of the MS.

9) Figures 3 and 4 is very difficult to read. Too small.

10) In Phylogenetic Analysis, please also focus on the issues of persimilis and nama, I don't recommend the authors to go too far in Nymphalidae, which is rather big and complex, sampling a few species for such a huge family is not adequate, please tune down the conclusion of this part.

11) Figure 11: The supporting values for a BI tree should between 0 to 1.

12) Again, please invite a native speaker to edit the MS, the quality of English language should be improved. 

Author Response

This manuscript is a resubmission of an earlier submission. The following is a list of the peer review reports and author responses from that submission.

Round 1

Reviewer 1 Report

The manuscript entitled “The complete mitochondrial genomes of Hestina persimilis, Hestina nama (Lepidoptera, Apaturinae) and taxonomic position of Hestina nama” has been evaluated. The merit of this work is sequenced two complete mitochondrial genomes of Apaturinae butterflies. However, the current version just like data paper, providing a few findings to scientific community. Especially, the description of mitochondrial genomes is as the same as other reports of mitogenome announcement. The most interesting part is to discuss the taxonomic position of “Hestina nama”. Unfortunately, the systematic treatments from the authors was just compared their phylogenetic position of the two mitogenomes, and no further systematic treatments have been careful revised. Some suggestions to authors if they would like to submit a revision on this journal.

  1. The most important goal is to clarify the monophyly of Hestina. The Authors should gather other published work on Apaturinae butterflies. I know Mr. Akio Masui has published many papers on this group, including “Hestinalis” nama. Therefore, your question can be resolved by other work.
  2. Stop the general description of mitochondrial genomes. There are too many data papers about mitogenomic issues described 37 genes, start/stop codons, and codon usage in their context. I suggest to describe “difference” from common insect mitochondrion (Boore, 1999). Start and stop codon are all prediction, not the real transcription data to support your results.
  3. Please give reader a good reason why the authors select Hestina persimilis and Hestinalis nama to discuss their phylogenetic position. Why not to sample others (Hestina nicevillei, Hestinalis mimetica, Hestinalis melanoides, Hestinalis namida, Hestinalis waterstratdi, Hestinalis divona, etc)?
  4. Ohshima et al (2010) have published an update work on Apaturinae butterflies, but authors consider their grouping is not correct. However, I find your data is concordant to their results.

Reviewer 2 Report

The present MS reports the mitogenomes of Hestina persimilis and H. nama and analyses the phylogenetic position of both species. Based on the analysis, the authors propose that H. nama belongs to a different genus and encourage further taxonomic studies. This MS, in its current form, bears some fatal problems and does not suitable for publication on Insects. I hereby offer two ways to the authors with my recommendation as Major Revision. If they wish to emphasise on mitogenome announcement, submitting this MS to other journals, such as Mitochondrial DNA Part B is a better choice. If they wish to answer the question of H. nama, the MS must be extensively revised with the supervision of a Nymphalidae taxonomist. Below are my comments.

First of all, the MS including its future versions must undergo language editing by a native speaker. The expression and wording of this version are problematic.

The taxonomy background of this MS is of extreme concern. I feel that the authors are not equipped with adequate taxonomy knowledge to understand the question surrounding Hestina and Hestinalis. Judging from the citations, they only cited Chou (1994) [should be Chou I., not Zhou Y.] and Lee & Zhu (1992) [should be Lee C. L., not Li C. L.], which are already outdated. Hestinalis nama was originally described as Diadema nama by Doubleday in 1844, on page 97 of the first volume of “List of the Specimens of Lepidopterous insects in The Collection of the British Museum”. Therefore, saying nama was firstly placed under Hestinalis and citing Lee & Zhu (1992) is wrong. Subsequently, other authors moved this species under genera Hestina and Hestinalis, which is the core debate of this MS. However, in many recent literatures, such as Bozano (2011) Guide to the Butterflies of the Palearctic Region - Apaturinae, and Lang (2012) The Nymphalidae of China, as well as the most updated Wu & Hsu (2017) Butterflies of China, Hestinalis is treated as a distinct genus. Therefore, the hypothesis of this MS is incorrect. The authors should refer to updated literature and rewrite the background. Morphologically, including genitalic structure, Hestina and Hestinalis are easily separable, sensu Bozano (2011). The absence of DNA does not affect our understanding of taxonomy of these taxa at all.

The meaning of using DNA is to validate taxonomy and solve debates. If the authors wish to reach this destination, they should focus on Hestina and Hesinalis, while not limit they taxon scope inside China. Please include as many taxa as possible globally and answer the phylogenetic and phylogeographic questions using mitogenomic data.

The conclusion is strange. Instead of answering the topic, the authors went rather far away on irrelevant issues, this bit must be edited also.

At last but not the least, I must remind the authors that solid taxonomy is the critical foundation of species-based studies, one does such research should always update taxonomic information, or invite taxonomist as co-authors. Relying on a very limited number of outdated literature to form conclusions is dangerous.

Minor comments about the tree: All species names should be in italic font, while keeping the accession numbers in regular.